# Cl-out is a novel cooperative optogenetic tool for extruding chloride from neurons

Hannah Alfonsa[1], Jeremy H. Lakey[2], Robert N. Lightowlers[2] & Andrew J. Trevelyan[1]

Chloride regulation affects brain function in many ways, for instance, by dictating the GABAergic reversal potential, and thereby influencing neuronal excitability and spike timing. Consistent with this, there is increasing evidence implicating chloride in a range of neurological conditions. Investigations about these conditions, though, are made difficult by the limited range of tools available to manipulate chloride levels. In particular, there has been no way to actively remove chloride from neurons; we now describe an optogenetic strategy, 'Cl-out', to do exactly this. Cl-out achieves its effect by the cooperative action of two different component opsins: the proton pump, Archaerhodopsin and a chloride channel opsin. The removal of chloride happens when both are activated together, using Archaerhodopsin as an optical voltage clamp to provide the driving force for chloride removal through the concurrently opened, chloride channels. We further show that this novel optogenetic strategy can reverse an *in vitro* epileptogenic phenotype.

---

[1] Institute of Neuroscience, Medical School, Framlington Place, Newcastle upon Tyne NE2 4HH, UK. [2] Institute for Cell and Molecular Biosciences, Medical School, Framlington Place, Newcastle upon Tyne NE2 4HH, UK. Correspondence and requests for materials should be addressed to A.J.T. (email: andrew.trevelyan@newcastle.ac.uk).

Normal brain function arises from a delicate, and continually fluctuating, interplay between excitatory and inhibitory synaptic forces, and is therefore very sensitive to anything that affects the balance between excitation and inhibition. Chloride regulation in neurons is particularly important in this respect, because the major forms of synaptic inhibition, including $GABA_A$, $GABA_C$ and glycine receptors, are chloride-conducting channels[1–3]. Consequently, changes in intraneuronal chloride levels can markedly alter the synaptic effects of both GABA and glycine. Chloride balance further influences neuronal function by modulating transmitter loading into vesicles[4,5], by affecting neuronal acid–base balance through interactions with bicarbonate[6,7], and cell volume through osmotic effects[8].

The level of intraneuronal chloride fluctuates widely both over time[9–11] and in different parts of the neuron[12,13], suggesting that this may play a key role in dictating the functional state of the neuron. Very low levels of intraneuronal chloride means that the GABAergic events are strongly hyperpolarizing, and the inhibitory effect is very strong. Raised intraneuronal chloride, on the other hand, arising either by excess chloride entry or reduced clearance, shifts the GABAergic reversal potential ($E_{GABA}$) in a positive direction, and GABA can even become excitatory if $E_{GABA}$ exceeds action potential threshold. Such chloride-dependent disinhibition is thought to be a major factor in several brain disorders[14–17], most notably epilepsy[18–21].

The relative expression levels of two different cation-chloride co-transporters, KCC2 and NKCC1 (ref. 22), as well as local impermeant anions are reported to influence the baseline chloride balance[23]. At other times, intense synaptic bombardment, especially when there is concurrent glutamatergic and GABAergic activation[24], can overwhelm the capacity of neurons to regulate chloride levels, and intraneuronal chloride rises sharply at these times[25,26].

The true extent of these physiological and pathological dynamic changes, however, is not known, partly because we lack the tools for manipulating chloride. Optogenetics has introduced many new tools for regulating neuronal function[27], including the ability to load chloride into neurons using the optogenetic chloride pump, Halorhodopsin[28]. We used this strategy to demonstrate how raised intraneuronal chloride leads to an acute increase in network excitability[29] and alterations in spike-timing, consistent with patterns of activity that have been recorded in epileptic animals[30,31]. Similarly, the specific KCC2 antagonist, VU0463271, also rapidly induces epileptiform discharges in brain slice preparations[32]. The influence of chloride balance on spike-timing, mediated through its effect on $GABA_A$ inhibition[29], also has implications for a wide range of neuronal activity, including spike-time dependent plasticity, neuronal clustering and information transfer through networks with weak glutamatergic synaptic connections.

Reducing intraneuronal chloride, on the other hand, has proved difficult. As such, we only really have the means to manipulate intracellular chloride in an upwards direction. If we were able to reduce intracellular chloride, this would not only facilitate investigations into chloride regulation and its consequences, it may also provide a means of treating the various neurological conditions associated with raised chloride. The diuretic, bumetanide has some efficacy in this regard, by blocking NKCC1, and has shown promising antiepileptic action in animal models[33]. Its effect, however, is likely to be weak, acting only on chloride entry and not promoting chloride extrusion at all. A recent clinical trial to repurpose bumetanide for treatment of childhood epilepsies was terminated early because of otolaryngeal side-effects[34]. Significantly though, there are no means currently to remove chloride from neurons. We therefore decided to create one using optogenetics (Fig. 1), by coupling the hyperpolarizing proton pump, Archaerhodopsin (Arch)[35], with one of the newly described optogenetic chloride channels[36,37]. We demonstrate the action of this new optogenetic strategy using dissociated neuronal cultures, and then provide a proof of principle of its network effect, using it to reverse a simple, *in vitro* model of epilepsy.

## Results

**Optogenetic manipulation of intraneuronal chloride levels.** We first considered trying to reverse the orientation of Halorhodopsin so that it pumps chloride out of cells, but this strategy is problematic because the period of chloride extrusion would be accompanied by intense depolarization of the neurons. A second strategy presented itself with the publication of two different optogenetic chloride channels, ChloC[36] and the anion channel rhodopsins (ACRs)[37]. We reasoned that if we could control the membrane potential, then we would also have control over the direction of chloride movement through these channnels. We demonstrated this proof of principle by performing gramicidin perforated patch clamp recordings of dissociated cultured neurons, expressing either ChloC or GtACR. This allows a non-invasive measurement of GABAergic reversal potentials induced by brief puffs of the $GABA_A$ agonist muscimol delivered from a nearby micropipette. We measured $E_{GABA}$ before and after activation of the optogenetic chloride channels, while holding the cell at a relatively depolarized, or a hyperpolarized level. A voltage ramp was delivered close to the timing of the peak GABAergic conductance. In both cases, $E_{GABA}$ shifted towards the holding membrane potential: when the cells were held at $-50\,mV$, there was a positive shift in $E_{GABA}$ for both ChloC (avg ± s.e.m.: $2.69 \pm 0.55\,mV$, $n = 6$) and GtACR ($7.93 \pm 1.38\,mV$, $n = 5$); when instead the cells were held at $-80\,mV$, there was a negative shift in $E_{GABA}$ (ChloC: $-1.93 \pm 0.61\,mV$, $n = 5$; GtACR2: $-3.43 \pm 0.64\,mV$, $n = 5$).

We then investigated whether the optogenetic proton pump, Arch, could provide the hyperpolarizing voltage clamp to drive chloride extrusion through a concurrently activated chloride channel. Since the effect depends critically on the Arch opsin altering the cell's membrane potential, these assessments could not be done using voltage clamp. Instead, we recorded cells in current clamp, again using gramicidin patch pipettes, and measured the shift in the inhibitory synaptic potential ('$GABA - \Delta V$'), induced by a brief puff of muscimol, as a proxy for a direct, voltage-clamp measure of $E_{GABA}$ (Fig. 1c).

Co-transfection strategies, using separate vectors for the two opsins, produced a mixed population with many neurons only expressing one of the opsins, and only some expressing both. Recordings of these doubly-transfected neurons suggested that this strategy could be successful. Critically, however, we showed that cells expressing ChloC or GtACR in isolation tended to produce a small positive shift in $GABA - \Delta V$ (Fig. 1d), so any transfection strategy resulting in networks where significant numbers of singly-transfected neurons expressing these channels are likely to produce deleterious effects, rather than improvements in network behaviour. We therefore incorporated the expression of both opsin elements into a single-expression system. We used two different co-expression methodologies linking two different pairs of opsins, to yield four different constructs in total (Fig. 2). The first co-expression method used a 'self-cleaving' T2A linker peptide, derived from *Thosea signa* virus, which allows two separated proteins to be expressed under a single promoter[38]. The second method used a β-HK linker sequence[39], to create a single-fusion protein by concatenating the two opsins. Because the function of these paired expression systems depend on the cooperative action of both opsin components, we term these new proteins 'co-opsins',

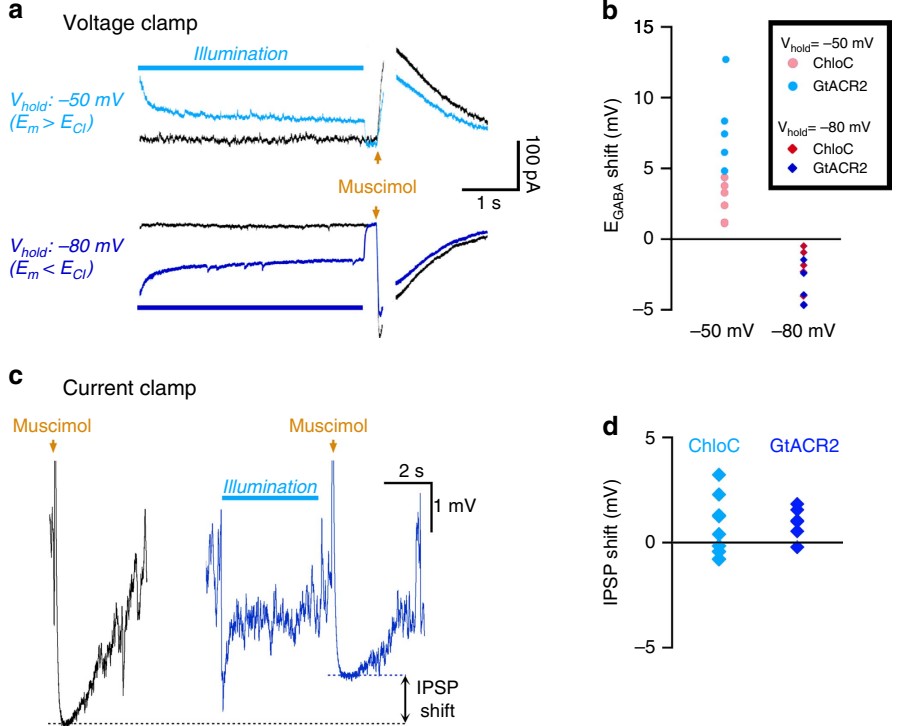

**Figure 1 | $E_{GABA}$ can be modulated, by using voltage clamp to direct chloride movement through ChloC or GtACR.** (**a**) Example traces of gramicidin patched neurons which express GtACR, showing control responses to a puff of the GABAA agonist, muscimol (black) when the cell is held at either −50 mV (upper) or −80 mV (lower). The blue traces are repeat trials in the same cells, with a prior activation of GtACR for 4 s, stopping 200 ms before the muscimol application. Voltage ramps were applied close to the peak of the muscimol response to determine $E_{GABA}$ more accurately (the ramps are excised from this figure so as to illustrate better the change in the IPSC shape. An example showing the ramps is in Supplementary figure 1). (**b**) Pooled data showing that the holding potential influences opsin-mediated shift in $E_{GABA}$, with positive shifts resulting when the cell is held at −50 mV, and negative when the cell is hyperpolarized (ChloC, $n = 6$ neurons; GrACR2, $n = 5$). (**c**) Example recording of a neuron transfected with GtACR2 only, illustrating the current clamp assay of the shift in $E_{GABA}$. (**d**) Group data for the shift in IPSP voltage following a 5 s opsin activation, in cells transfected either with GtACR2 ($n = 6$) or ChloC ($n = 8$).

and this specific co-opsin, we have called 'Cl-out', named for its cellular effect of removing chloride from the cytosol.

**Co-opsin expression and function.** The constructs produced good membrane localization (Supplementary figure 2), without affecting either baseline input resistance (Cl-out4 constructs; non-transfected cells, $n = 5$, $R_N = 1.30 \pm 0.06$ MΩ; transfected cells, $n = 5$, $R_N = 1.74 \pm 0.26$ MΩ; p = 0.176, NS.), or $E_{GABA}$ (non-transfected cells, $n = 5$, $E_{GABA} = -51.8 \pm 2.8$ mV; transfected cells, $n = 5$, $E_{GABA} = -47.2 \pm 2.8$ mV; $P = 0.249$, NS). Importantly, the distinct functions of both individual optogenetic elements were readily seen in whole-cell patch clamp recordings using a high chloride electrode filling solution (Fig. 2b). Blue (488 nm) illumination activates preferentially the chloride channel, producing a strong depolarization in cells filled with chloride, whereas green illumination (561 nm) preferentially activates Arch leading to a hyperpolarization. When both opsins were simultaneously activated, in cells recorded through gramicidin perforated patch clamp, it induced a rapid shift in shape of the muscimol-triggered post-synaptic potential (Fig. 2c-i). Notably some recordings even showed a reversal of the sign of the post-synaptic potential (see Fig. 4a, lower trace), demonstrating that the change is unlikely to be explained by a change in conductance, but rather by a change in $E_{GABA}$. This was further confirmed by voltage-clamp ramp recordings, showing no change in the slope (Supplementary figure 3; percentage conductance change $= -3.5 \pm 2.5\%$, $n = 3$). We concluded therefore that Cl-out activation produces its effect on GABAergic

events by reducing intracellular chloride levels. In contrast, activation of the individual opsins singly had little or no effect on subsequent GABAergic events (Fig. 1c, Fig. 2c-ii).

We assessed the chloride removal by measuring the $GABA - \Delta V$ induced by opsin activation (5 s illumination) with the two different Cl-out constructs incorporating either ChloC (Cl-out1 and 3) or GtACR (Cl-out2 and 4; Fig. 3). We failed to get good expression of Cl-out3, but all other co-opsins induced significant negative $GABA - \Delta V$, and if cells were preloaded with chloride by blocking KCC2 using VU0463271 (10 μM; baseline $GABA_{peak} = -74.7 \pm 3.3$ mV, $n = 9$; VU0463271 $GABA_{peak} = -55.9 \pm 3.0$ mV, $n = 6$; $P < 0.005$), the negative shift was significantly increased over what was possible from baseline (baseline $GABA - \Delta V = -1.89 \pm 0.33$ mV; $GABA - \Delta V$ with VU0463271 $= -4.19 \pm 0.58$ mV, $P < 0.005$ $t$-test; Fig. 3), indicating that concentration gradient and leakage does impose an upper limit on the degree of chloride extrusion.

Following the end of the illumination, the neurons slowly accumulated chloride once more, but with a substantially longer time constant (VU0463271 bathed neurons, $\tau = 37.5 \pm 6.7$ s; $n = 7$; Fig. 4) than of previous measures of the correction of optogenetically altered chloride levels ($\tau = 8$–12 s)[28,29]. In cells bathed additionally in bumetanide (10 μM to block NKCC1), the recovery was significantly slower ($\tau = 58.2 \pm 11.5$ s; $n = 7$; $P < 0.025$; Fig. 4b,c), an effect that was reversed by washing out bumetanide ($\tau = 33.1 \pm 5.1$ s; $n = 5$). These results suggest both NKCC1 and KCC2 contribute significantly to chloride movements across the cell membrane at rest.

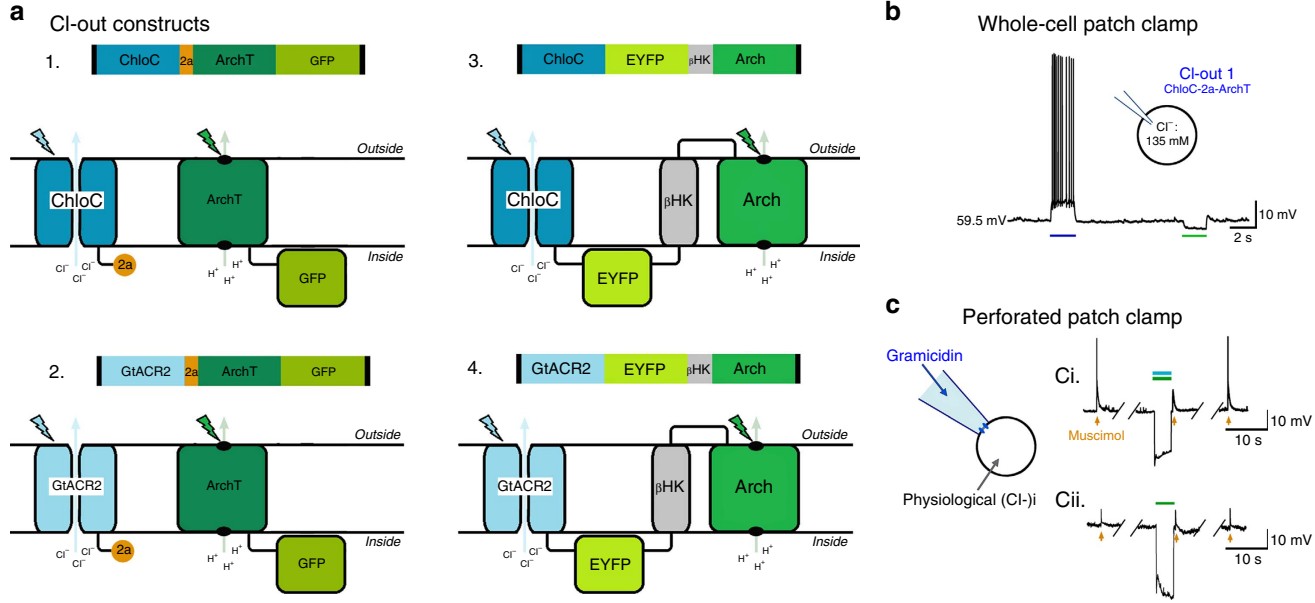

**Figure 2 | Co-expression strategies to achieve independent function of Arch with either ChloC or GtACR.** (**a**) Schematics of four different Cl-out constructs, using either ChloC (Cl-out1 and 3) or GtACR (Cl-out2 and 4) as the chloride channel, linking them either with a T2A protein (Cl-out2 and 3) or with a βHK linker sequence (Cl-out3 and 4). (**b**) Demonstration of the independent action of the two opsins in a pyramidal cell recorded in a brain slice prepared from a young adult mouse, following transfection by injection of a viral vector intracortically at postnatal day 1. The electrode filling solution had a high Cl content, meaning that $E_{Cl}$ was close to 0 mV, and so opening of the ChloC-component-triggered action potentials, while activation of the Arch component hyperpolarized the cell. (**c**) Perforated patch clamp recording of transfected cells to assess shifts in $E_{GABA}$ by monitoring the response to the GABAergic agonist, muscimol. The upper recording (**c**-i), of a cell transfected with Cl-out1, shows that activation of both opsin components (488/561 nm; upper trace) resulted in a marked reduction in the amplitude of the GABA$_A$ event reflecting a large negative shift in $E_{Cl}$. This particular example, for illustrative purposes, was bathed in the KCC2 blocker, VU0463271, resulting in enhanced baseline intracellular Cl$^−$, and thus a bigger effect of Cl-out1 activation. In contrast, (**c**-ii) in a cell transfected only with Arch, opsin activation (lower trace; 561 nm, green) did not alter the response to muscimol.

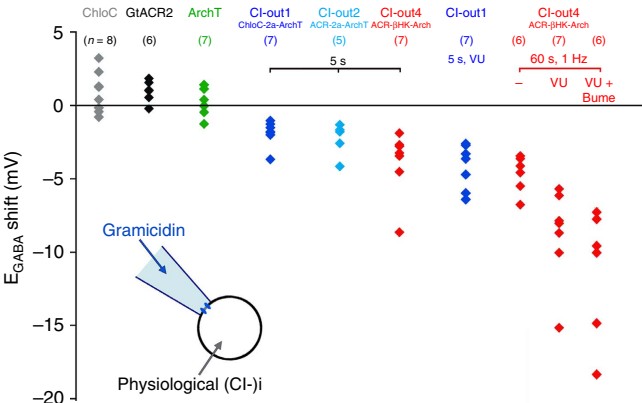

**Figure 3 | Pooled data showing the effect of activation of different opsins on $E_{GABA}$.** The data show the shift in IPSP peak (muscimol puff, see Fig. 2) following a brief, opsin activation (5 s continuous activation unless otherwise indicated), measured in perforated patch recordings of dissociated cultured neurons. In some cases, cells were preloaded with chloride by blocking KCC2 (using VU0463271, 'VU') and NKCC1 (bumetanide, 'Bume'). Numbers of cells recorded in each set are given in brackets.

## Cl-out can persistently reduce network excitability. We next examined whether we could use this approach to modulate the activity patterns in large networks of neurons simultaneously (Fig. 5). We incorporated the Cl-out1 and Cl-out4 constructs under the pan-neuronal, hSyn promoter, together with an enhanced yellow fluorescent protein (EYFP) marker sequence, into adeno-associated viral vectors, and injected the virus into the

left cerebral ventricle of newborn mouse pups. Both vectors achieved *in vivo* expression, but levels of Cl-out1 were far better than for Cl-out4, so subsequent network modulation experiments were performed only for Cl-out1 constructs. Animals were allowed to grow to maturity, and at between postnatal 2–4 months age, they were killed to prepare coronal brain slices. We selected brain slices showing evidence of significant EYFP labelling in the hippocampal formation, and recorded extracellular activity using Neuronexus probes in the CA1 pyramidal layer. We recorded baseline periods, before introducing the KCC2 blocker, VU0463271 (10 μM) into the bathing media. As reported previously[32], VU0463271 perfusion induced a marked increase in the network excitability, with the appearance of large spontaneous network discharges and also a much larger response to electrical stimulation (Fig. 5b). We then delivered a series of 25 s illuminations, at 30 s intervals (83% duty cycle), using both blue (488 nm) and green light (561 nm) to activate the expressed Cl-out1 co-opsin. We tested network excitability in each cycle of illumination, during the 5 s dark period, following the same protocol described previously[29]. In all cases, this rapidly reversed the hyperexcitable state back to baseline levels, as measured both by the near abolition of spontaneous events (Fig. 5c; normalized frequency following light activation = 15.3 ± 2.8%; highly significant difference from 100%, $P \ll 0.001$) and the intensity of the responses to electrical stimulation (Fig. 5d; Normalized power, VU, 38.4 ± 12.4; VU + light, 6.1 ± 2.1; $P < 0.05$), even in the continued presence of perfusing VU0463271.

We next asked whether these effects could be achieved by activating the individual component opsins (Fig. 6). For the ArchT experiments, we used animals transfected only with that construct (Fig. 6a). For the ChloC experiments, we used Cl-out1 transfections, utilizing the separation of spectral sensitivities of

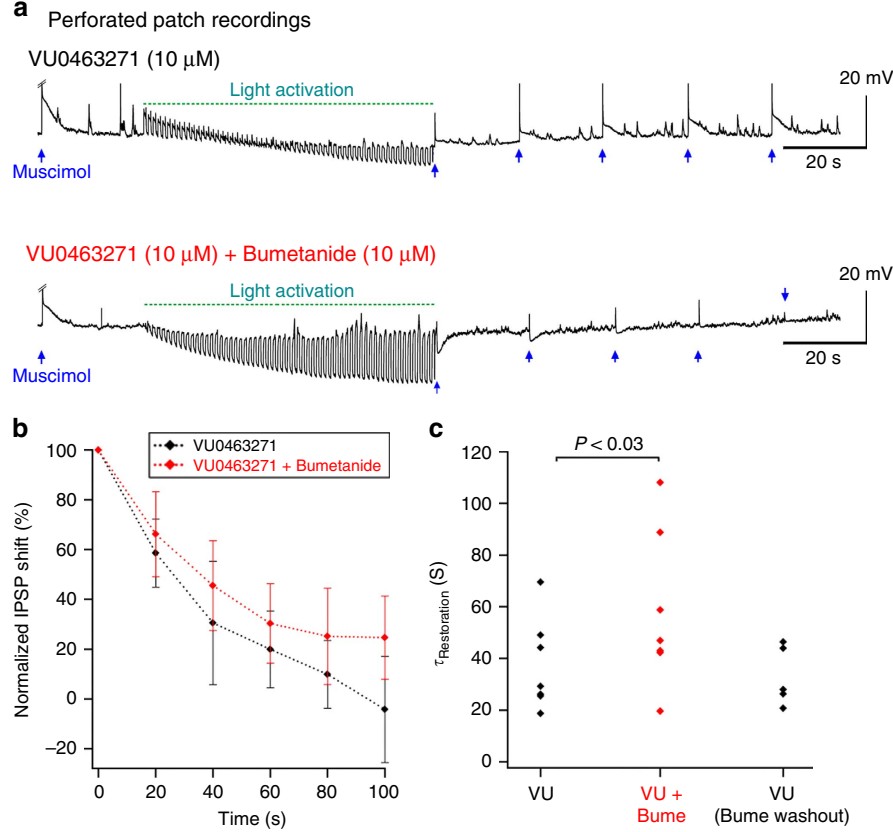

**Figure 4 | Persistent shifts in $E_{GABA}$ induced by Cl-out4 activation in cells with compromised cation-chloride co-transport.** (**a**) Example recordings showing repeated muscimol applications, either side of a period of intense Cl-out activation, in cells bathed in VU0463271 and bumetanide to block KCC2 and NKCC1 respectively. A test muscimol application was delivered every 20 s, because the response was stable at that frequency in baseline conditions. Note the progressive shift in the voltage response during the reiterative light activation, reflecting the shifting ChloC/GtACR component of the current as $E_{Cl}$ shifts more negative. (**b**) Pooled data of the IPSP shift, normalized to the maximal shift, immediately after the period of opsin activation (mean ± s.e.m.). (**c**) Pooled data of the time constant for recovery of GABA − ΔV to baseline levels, showing a significant additional slowing of recovery when both KCC2 and NKCC1 were blocked ($\tau_{(VU + Bume)} = 58.2 \pm 11.5$ s, $n = 7$), over the situation where only KCC2 was blocked ($\tau_{(VU)} = 37.5 \pm 6.7$ s, $n = 7$; $\tau_{(VU\ (Bume\ washout))} = 33.1 \pm 5.1$ s, $n = 5$).

the two opsins (see Fig. 2b), and contrasted the effects of illuminating only with blue light with combined blue/green illumination (Fig. 6b). For both opsins individually, the intensity of the response to electrical stimulation following illumination tended to increase, and substantially so in two of the ArchT experiments (Fig. 6c). These experiments confirmed that the restorative action of Cl-out1 in reducing network excitability was indeed due to the cooperative action of the two component opsins.

Once the illumination was terminated, the hyperexcitable state was re-established, but only slowly, over many tens of seconds (example traces in Figs 5b and 6b; pooled data in Supplementary figure 4), consistent with the cellular time constants shown in Fig. 4. This shows that the inhibitory benefit of Cl-out far outlasts its activation, consistent with our cellular measures (Fig. 4). This property represents a significant advance in our optogenetic, antiepileptic armoury because it is directed at reinvigorating the brain's own endogenous protective mechanism, which previous work has shown to be extremely powerful[40,41], an effect that long outlasts the period of optogenetic activation.

Finally, we examined whether manipulating intraneuronal chloride levels in this way could influence spike timing, because there is extensive evidence that spike time is strongly regulated by $GABA_A$ activation[42,43]. We tested, using multiunit recordings, whether Cl-out activation changed the entrainment of spiking to the dominant field oscillation, in the opposite direction to the

effect shown previously, when we drove chloride into neurons using Halorhodopsin[29]. We first applied VU0463271, to load chloride generally into neurons, and then assessed spike entrainment before and after Cl-out activation. These experiments showed a marked sharpening of the entrainment of spikes, as indicated by a significant reduction in the half-width index, which is a measure of the skew of the histogram towards a single peak in the oscillation. This reduction only occurred with co-activation of both opsins together, and not with activation of either ArchT or ChloC alone (Supplementary figure 5).

## Discussion

We have developed a novel approach to using optogenetics, in which two opsins are used cooperatively to achieve the desired alteration of neuronal function. Whilst others have expressed different optogenetic proteins in single cells before[39], these previous studies have done so to provide different optogenetic effects brought about by different colour light; their aim was always to activate one or other protein, and not both together, within the constraints offered by their spectral sensitivities. In contrast, the effect we sought was dependent on the cooperative action of both optogenetic elements. This is the first utilization of optogenetics to provide a voltage-clamp effect to move ions unidirectionally through a channel, and in this way, provides the first means to actively extrude chloride from neurons.

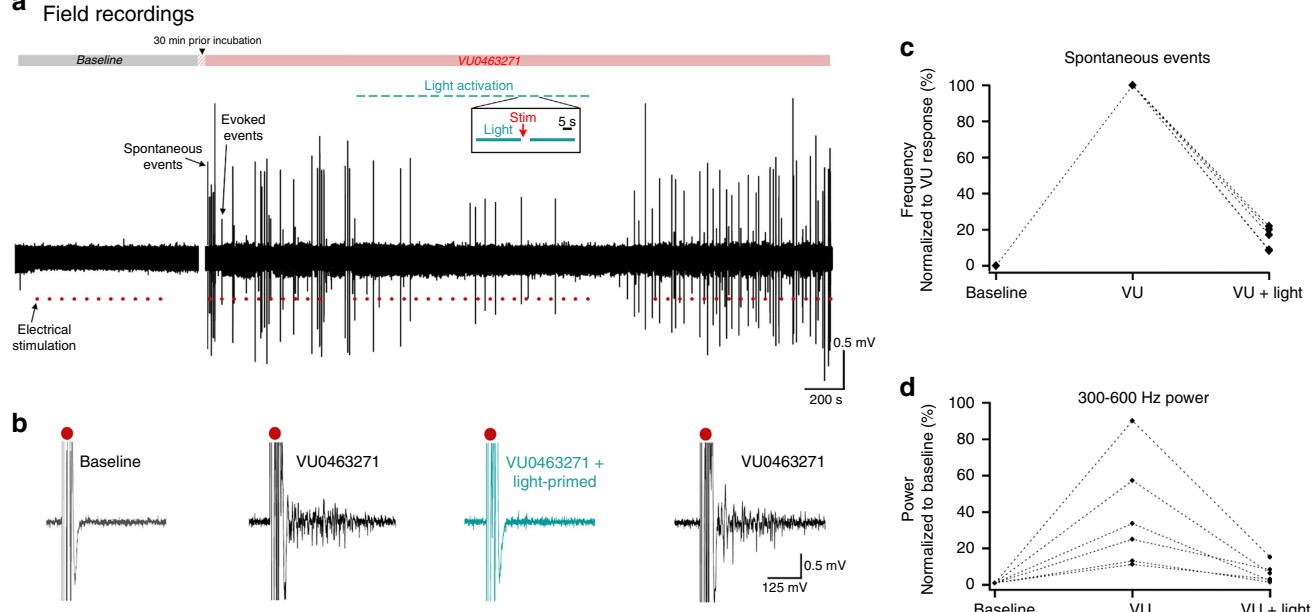

**Figure 5 | Reversal of excitable network state by Cl-out1 activation.** (**a**) Field recordings in six brain slices prepared from six young adult mice transfected with Cl-out1 (injected AAV viral vector on the day of birth). Epileptiform activity was induced by 30 min prior incubation in VU0463271. Red dots indicate times of electrical stimulation, and expanded sample events are shown below (**b**). The VU-induced excitability was reversed consistently by a period of Cl-out activation, performed intermittently (25 s light on, 5 s light off), with the electrical stimulation occurring in the dark period of the cycle. There were highly significant reductions both of spontaneous events (**c**) and the multiunit bursts in response to electrical stimuli, assessed by the power in the 300–600 Hz frequency band (**d**).

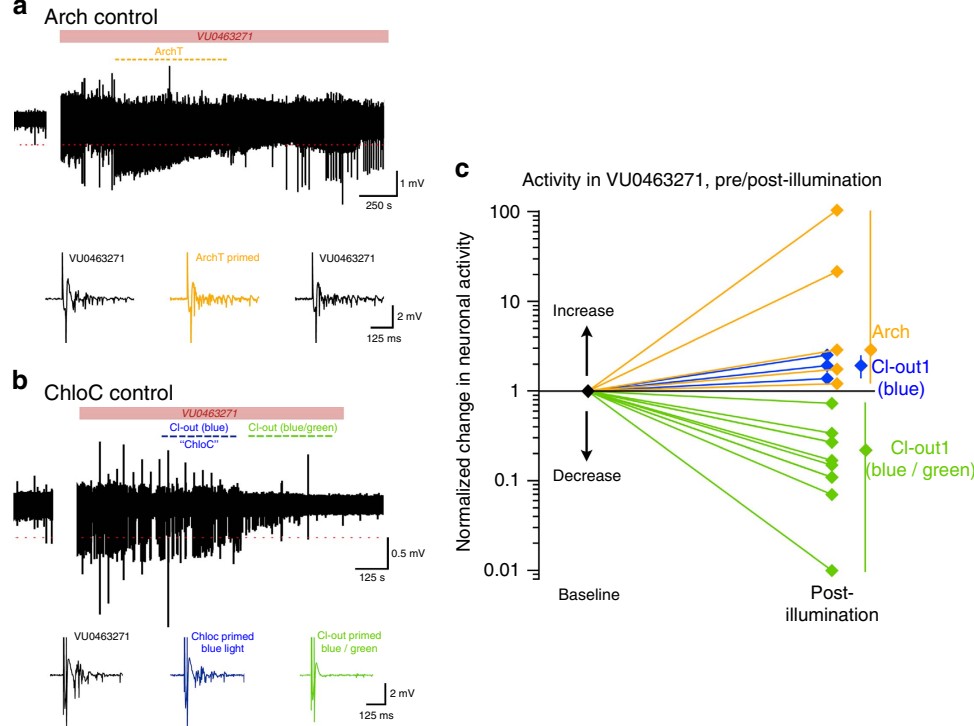

**Figure 6 | Contrasting effects of Cl-out1 activation with the effects of activation of either opsin individually.** (**a**) Example field recording from CA1 of an animal injected with AAV viral vector for ArchT. Epileptiform discharges were induced by bathing the brain slice in VU0463271, and 10 cycles of ArchT activation (25 s on, 5 s off) were made using illumination at 561 nm (solid-state laser light delivered through an optic fibre). (**b**) Field recording of a Cl-out1 transfected CA1 region, illuminated first with blue light (epifluorescent illumination through 4 × air objective) to activate the ChloC opsin specifically, followed by illumination with both blue and green light (561 nm). (**c**) Pooled data showing the change in the response to electrical stimulation induced by opsin activation, relative to the pre-illumination state (ArchT, 5 slices from 2 animals; ChloC, 3 slices from 2 animals; Cl-out, 8 slices from 6 animals). The range and mean values are shown to the right. The baseline state, to which the data were normalized, was the pre-illumination tissue bathed in VU0463271.

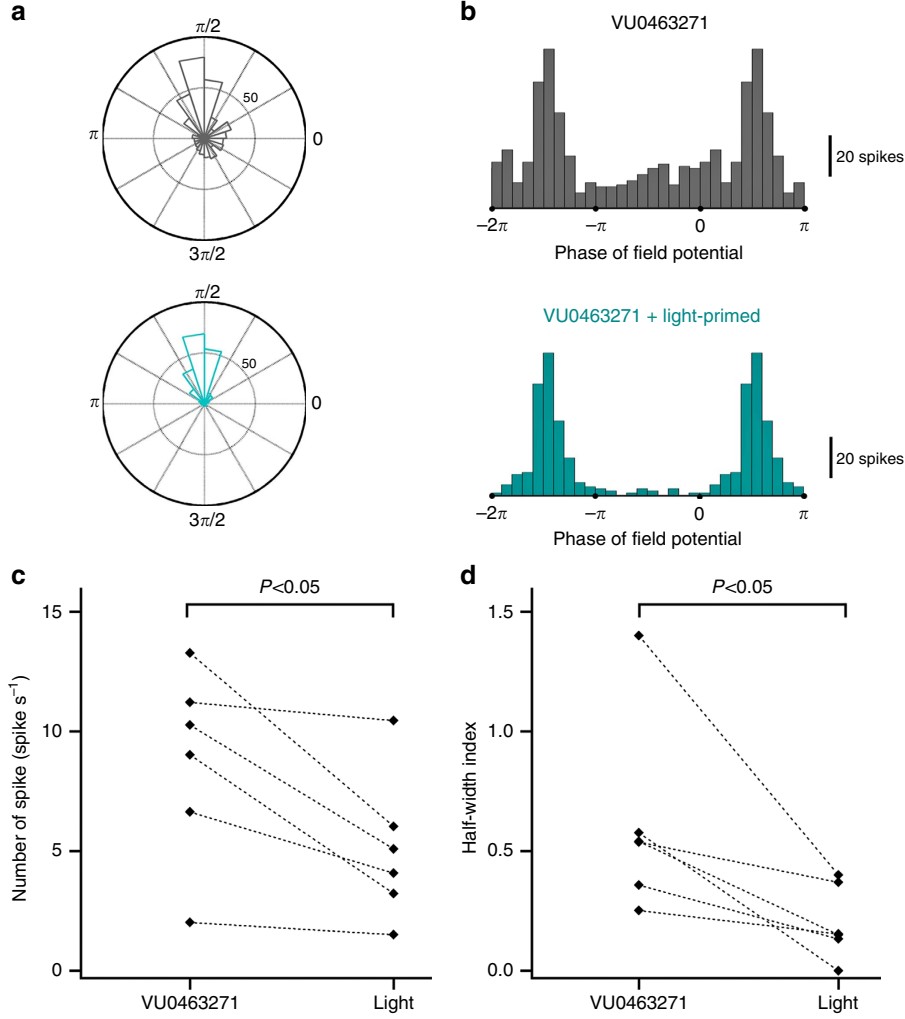

**Figure 7 | Altered spike-timing following Cl-out1 activation. (a)** Rose plot derived from one example recording, indicating the times of action potential spikes relative to the dominant local field oscillation. **(b)** The same data shown as histograms of the different phases of the cycle (note that the bins from $-2\pi$ to $-\pi$ are replicated ($0-\pi$)). Pooled data (6 slices from 4 animals) showed significant shifts in both the number of spikes **(c)** and the half-width index **(d)**, which is a measure of the entrainment of the spiking to the local field potential[29].

The dependence of the Cl-out effect on Arch pumping a proton out of the cell may conceivably alter the pH balance slightly, increasing the intracellular pH relative to outside. This, though, is likely to be minimized by the presence of intracellular carbonic anhydrase and the extracellular pH buffers. It is important to note, however, that such a pH shift would make the bicarbonate reversal potential marginally more positive (that is, the opposite of the $E_{GABA}$ shift we actually measured), meaning that the change we measure cannot be ascribed to any pH change, a point that is further proved by our control experiments using Arch activation alone.

Our development complements our recent demonstration that chloride can be driven into neurons acutely using Halorhodopsin[28,29], so that now we have the facility for bidirectional control over chloride levels. Currently, this could not be achieved in the same cells, because the spectral activation curves of Halorhodopsin and the Arch component of the Cl-out constructs overlap. So the potential is only realized for one strategy in a given cell, but the co-opsin strategy that we describe here will lend itself to other combinations of opsins with shifted spectral sensitivities as they become available. In this regard, it should also be noted that combining GtACR1 with ArchT might

yield a co-opsin which could be activated by a single wavelength, but the rather slow kinetics of GtACR1 would then present other problems regarding the temporal matching of the co-opsin effect. Research using these new tools for manipulating chloride will also benefit from developments in imaging chloride levels in cells[44–46], and has the potential to usher in a new era of research into the role of chloride in neurons. As we have mentioned, chloride levels are highly variable in different cells and across different times, and given what we know about the consequences for neuronal function, it seems highly likely that chloride regulation is an important mechanism for dictating neuronal, network and brain states. In particular, by altering the strength of GABAergic control, changes in intraneuronal chloride is hypothesized to influence spike timing and jitter, and this theory can now be tested. Our new tool also has the potential to correct chloride loading in neurons, a common feature of epileptic networks. Accordingly, we have provided a demonstration that Cl-out can reverse an epileptic phenotype *in vitro*, and also alter spike-timing, paving the way for future explorations of its effect on various spike-time dependent mechanisms, and also on pathological processes, in more realistic animal models and potentially ultimately in man.

## Methods

**Animal Regulation.** All animal handling and experimentation were done according to UK Home Office guidelines, according to the requirements of the United Kingdom Animals (Scientific Procedures) Act 1986.

**rAAV cloning and production.** The different vectors were based on pAAV-hSyn-ChloC-2A-tDimer (purchased from Addgene #52455). To construct pAAV-hSyn-ChloC-2a-ArchT-GFP (Cl-out 1), ArchT-GFP was isolated from pAAV-CAG-ArchT-GFP (purchased from Addgene #29777) and subcloned into pAAV-hSyn-ChloC-2A-tDimer after removal of tDimer component using the BamHI and HindIII restriction sites. For the linked protein (pAAV-hSyn-ChloC-EYFP-bHK-Arch), EYFP-bHK-Arch cDNA was custom synthesized by GeneArt Company and subcloned into the vector using KpnI and HindIII restriction sites, excluding the 2A component. GtACR2 cDNA was then synthesized by GeneArt. It was subcloned into the pAAV-hSyn-ChloC-2a-ArchT-GFP and pAAV-hSyn-ChloC-EYFP-bHK-Arch vectors to replace ChloC, using EcoRI and KpnI restriction sites. Sequence confirmation was outsourced (GATC Biotech). All vectors were amplified using recombinant deficient strain of E.Coli (OneShot Stbl3, Invitrogen) and purified using EndoFree Maxi Kit (QIAGEN #12362). Viral vectors were packaged as AAV serotype 2 by University of North Carolina vector core.

**Dissociated neuronal culture.** Assessment of the chloride-extrusion effect of the different proteins was performed using dissociated neuronal cultures. Primary neuronal cultures were prepared from rat pups at embryonic day 18–20 in the following way. A pregnant Sprague–Dawley rat was killed by cervical dislocation and a sagittal incision was made in the abdominal area to remove the pups. The neocortex and hippocampal tissue was isolated from the pups, and digested using papain enzyme (Sigma Aldrich) for 40 min. Cells were then dissociated within the growing medium (Neurobasal A, 2% B-27 supplement, 1% of fetal bovine serum, 0.5% glutamate, and 0.5% antibiotic–antimicotic, GIBCO Invitrogen) by gentle agitation using a pipette gun with a 10 ml tip. Transfection was done by electroporation using Rat Neuron Nucleofactor kit (Lonza Amaxa). For each electroporation, ∼5 million cells were collected by centrifugation (80 relative centrifugal force) for 5 min. Cell pellets were re-suspended in 100 µl nucleofactor solution and mixed with 1 µg of plasmid DNA. Cells were electroporated using an electroporation machine (Amaxa; programme G-013). Cells were then plated on pre-coated coverslips using poly-L-lysine with 1:10 dilution. Medium was changed 3 and 24 h post-plating. All experiments were performed after DIV 10.

**Patch clamp experiments.** Recordings were made using one of two laser spinning disc confocal microscopes (Visitech; Olympus) fitted with Patchstar micromanipulators (Scientifica) mounted on a Scientifica movable top plate. Electrophysiological data were collected using a Multiclamp 700B amplifier (Molecular Devices) and Digidata acquisition boards connected to Dell desktop computers running pClamp software (Molecular Devices). During the entire recording, cells were bathed in circulating oxygenated (95% $O_2$/5% $CO_2$) artificial cerebrospinal fluid (ACSF) solution (in mM: 125 NaCl, 26 $NaHCO_3$, 10 glucose, 3.5 KCl, 1.26 $NaH_2PO_4$, 1.2 $CaCl_2$ and 1 $MgCl_2$; perfusion at 1–3 ml min$^{-1}$) heated to 33–37 °C by a sleeve heater element (Warner Instruments) around the inflow tube.

**Whole-cell patch.** Whole-cell recordings were made using 3–7 MΩ pipettes made of borosilicate glass (Harvard apparatus). Electrode filling solution used was either low Cl$^-$ (K-methyl-SO$_4$ 125 mM, Hepes 10 mM, Mg-ATP 2.5 mM, NaCl 6 mM; 290 mOsM and pH 7.35) or high Cl$^-$ (KCl 135 mM, Na$_2$ATP 4 mM, Na$_3$GTP 0.3 mM, MgCl$_2$ 2 mM, and Hepes 10 mM; 290 mOsM and pH 7.35).

**Perforated gramicidin patch.** Gramicidin perforated patch recordings were made using 3–7 MΩ pipettes (borosilicate glass; Harvard Apparatus) filled with a high Cl$^-$ electrode filling solution. Fresh Gramicidin (Sigma Aldrich, G5002) stock was made daily with 5 mg ml$^{-1}$ concentration in dimethylsulfoxide. Gramicidin stock was added to the high Cl$^-$ electrode filling solution to achieve a final concentration of 0.1 mg ml$^{-1}$ and mixed by 40 s vortexing and 5 s sonication. Gramicidin electrode filling solution was then filtered using a 0.45 millex pore filter and used for patching straight away. Experiments were performed when series resistance had stabilized below 100 MΩ, which was typically achieved ∼30 mins after patching. The series resistance was required to be stable (<10% fluctuation) during the time course of data collection.

For voltage-clamp measurements of E$_{GABA}$ (Fig. 1a), recordings were made in voltage-clamp mode, with repeated 200 ms duration test ramps (triangular up–down function; peak − 50 mV; trough − 90 mV; slope ± 400 mV s$^{-1}$) applied at baseline and also at close to the peak of GABAergic current (I$_{GABA}$). All other assays of E$_{GABA}$ were made in current clamp mode, because we were assessing functionality of different Cl-out constructs in which the effect is critically dependent on a voltage shift provided by Arch. For this reason, voltage clamping the cells would have abolished the effect. We therefore measured the average shift in GABAergic inhibitory post-synaptic potentials (IPSPs) induced by muscimol (100 µM) application delivered close to the recorded neuron, from a patch pipette

coupled to a picospritzer (Parker Instrumentation, 10 ms pressure pulses,10–20 PSI)(see Fig. 1c). IPSPs were measured in the presence of 2 mM kynurenic acid, to block all glutamatergic synaptic events. The timing of the puff was coordinated with the illumination (optogenetic activation) via the pClamp software coupled to a Digitimer box. The shift in IPSP was calculated from averages of the membrane potential over 200 ms epochs, centred on the peak of the muscimol induced conductance, for pairs of events before and after activation of the Cl-out constructs. This measure reflects the shift in E$_{GABA}$, but is not a direct measure, and so we refer to it as GABA − $\Delta V$, to make this distinction clear. If action potentials occurred, these were curtailed at the peak of GABA event, before the measurement.

**Optogenetic expression.** For general neuronal expression of the optogenetic proteins, viral vectors (pAAV-hSyn-ChloC-2a-AchT-GFP manufactured by UNC viral vector core using plasmid constructs provided by ourselves; rAAV5/CAG-ArchT-TdTomato (UNC)) were injected into wild-type mice (C57BL/6 J). Injections were made into postnatal 0-day-old pups. The pups had local anaesthetic (EMLA, AstraZeneca) cream applied to the left top of their head, and were anesthetized subsequently using isofluorane inhalation. A single injection of virus was made using a 10 µL Hamilton syringe, with a bevelled 36 gauge needle (World Precision Instruments) ∼1 mm anterior to lambda, and 1 mm lateral to the midline into the left hemisphere at 1.7–0.8 mm deep to the skin (four separate 200 nl injections, deepest first). Approximately 0.8 µl (∼$10^{11}$–$10^{12}$ viral particles) was injected over a 12 min period. Animals were allowed to recover. Subsequent electrophysiological recordings were made between the ages of 5 weeks to 3 months.

**Brain slice experiments.** Coronal brain slices (350 µm for patch clamp experiments or 400 µm for network activity experiments) were prepared from the injected animals, at age 2–4 months (male), on ice-cold oxygenated (95% $O_2$/5% $CO_2$) artificial cerebrospinal fluid (ACSF: 125 mM NaCl, 26 mM NaHCO$_3$, 10 mM glucose, 3.5 mM KCl, 1.26 mM NaH$_2$PO$_4$, 3 mM MgCl$_2$). After cutting, the slices were transferred to incubation, submerged or interface chamber (room temperature) perfused with oxygenated ACSF (ACSF: 125 mM NaCl, 26 mM NaHCO$_3$, 10 mM glucose, 3.5 mM KCl, 1.26 mM NaH$_2$PO$_4$, 2 mM CaCl$_2$, 1 mM MgCl$_2$) for at least 1 hour before transfer to a recording interface chamber (33–36°C), after which time they perfused with slightly lower Ca$^{2+}$ ACSF (1.2 mM CaCl$_2$; ACSF otherwise as before). The divalent cation concentration followed the protocol of Sanchez-Vivez and McCormick (1999). In some experiments, cation-chloride cotransporters were blocked using 10 µM of VU0463271 (Tocris) to block KCC2, and 10 µM bumetanide (Tocris) to block NaKCC1 (minimum drug incubation time of 20 min).

Extracellular recordings were made using tetrodes in a diagonal arrangement separated by 50–100 µm (Neuronexus), connected to a 1401–5 Analog-Digital converter (Cambridge Electronic Design, Cambridge) and using Spike2 software (Cambridge Electronic Design) running on a desktop computer (Dell). Multichannel extracellular recordings were digitized at 20 kHz for high frequency bandpass (300–1,000 Hz) and 5 kHz for low-frequency bandpass (1–300 Hz), using 16 linear 4-channel-probes in diamond configuration (Q1x1-tet 'tetrode'; NeuroNexus) connected to a ME16-FAI-µPA-system and MC_Rack software (Multichannel Systems, Reutlingen). The tetrodes were mounted on patchmaster micromanipulators, and recordings were made from the pyramidal cell layer in CA1. Bipolar electrical stimulation (0.1 ms duration) was delivered via tungsten electrodes onto the white matter. Optogenetic proteins were activated by 561 nm, 50 mW solid-state laser (Cobolt) connected to a fibre optic with a fine cannula (400 µm core, 0.20 NA; Thorlabs; after optimizing the optic fibre coupling, the unfiltered light at the cannula tip was measured at 15–30 mW) synchronous with blue epi-fluorescence illumination (460 ± 20 nm excitation filter) controlled using an electrically-coupled, mechanical shutter (Sutter Instruments), and delivered through a 4 × air objective (0.28 NA, Nikon). The central illumination of the epifluorescent blue illumination was ∼10 mW mm$^{-2}$.

**Code availabilty.** Analyses were done using in-house software (available on request) implemented on either Igor (Wavemetrics) or Matlab. Sample sizes were chosen based on the large effect sizes found in our previous studies using Halorhodopsin to increase intracellular Cl$^-$ (ref. 29). Recordings and analyses were not blinded.

**Data availability.** The data that support the findings of this study are available from the corresponding author on reasonable request.

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

## Acknowledgements

We thank Gimmi Ratto, Claudia Racca, Andrei Sorin Ilie, Rolando Berlinguer-Palmini, Neela Codadu and Ryley Parrish for discussions and technical assistance at different stages of the project. Financial support for this project came from Newcastle University and MRC (Confidence in Concept grant).

## Author contributions

H.A. and A.J.T. designed experiments and co-authored the paper, performed viral injections and culture work. H.A. performed all recordings and analyses, and performed genetic engineering of plasmids. J.H.L. and R.N.L. consulted on molecular biological manipulations and provided training and facilities in these techniques. There are no conflicts of interest to declare.

## Additional information

**Competing financial interests:** The authors declare no competing financial interests.

