## [Peer Review File · Nature Communications]

Reviewers' comments:

Reviewer #1 (Remarks to the Author):

Alfonsa et al. develop and present a novel strategy for optogenetic Cl⁻ extrusion termed 'Cl⁻-out'. This ingenious technique utilizes the co-operative activity of two separate opsins, an archeorhodopsin, to hyperpolarize the membrane potential and generate the driving force for Cl⁻ efflux via the second opsin, a light-activatable chloride channel, either Chloc or gtACR2. The authors utilize gramicidin perforated patch-clamp recordings to convincingly demonstrate the ability of various Cl⁻-out constructs to lower intraneuronal Cl⁻ in dissociated culture systems. Furthermore, they demonstrate Cl⁻-out's ability to reduce epileptiform activity and spike rates in VU0463271 treated acute slices.

This is an original piece of work of considerable interest to the field as intracellular Cl⁻ regulates the strength and direction of GABAergic signaling. Interest in Cl⁻ homeostasis has recently seen a surge in popularity making this new tool a timely and welcome contribution.

I have one major comment and several minor comments:

Major comment:

Cl⁻-out's major novelty is its function as a Cl⁻ extruder, which the authors show in dissociated systems. However, it is important that the authors convincingly show this tool working as a Cl⁻ extruder in more intact preparations (ie acute slices) to demonstrate its utility for exploring how changes in Cl⁻ concentration relate to network states. I remain unconvinced that the presented effects (ie Fig 5 and 6) on network activity are definitely due to changes in Cl⁻ concentration and GABAergic signaling rather than direct voltage (Arch) or shunting (Chloc) inhibitory effects. Ie the effects described on epileptiform activity and spike timing relative to the dominant field oscillation (particularly during light illumination) could possibly be explained purely by the hyperpolarizing affect of Arch activation alone. I believe the authors need to follow the same route as in their 2015 paper using eNpHR. ie

- 1) Use a Cl⁻-out 'priming' approach. Ie analyze evoked and spontaneous activity during the 5s that the light is off (ie following the 25s light activation). Fig 4c demonstrates that the tau for Cl⁻ recovery in VU is 20-80s so Cl⁻ should be reduced in this window. This would allow for a dissection of the Arch/Chloc voltage/conductance effects vs the pure Cl⁻ concentration effects on network activity and spike timing relative to the field oscillation. It is worth noting that the authors state on pg 7 'Once the illumination was terminated, the hyperexcitable state was re-

established, but not instantly, indicating that the inhibitory benefit of Cl-out far outlasts its activation, consistent with our cellular measures' - this is promising but no data is presented to support this claim.

2) It would be useful (but not essential) to perform some gramicidin perforated patch-clamp recordings in acute slices and directly demonstrate that a Cl-out is able to lower intraneuronal Cl-.

Minor comments:

1) The Cl-out constructs are relatively large, and could conceivably result in cell trafficking / aggregation issues. If available could the authors provide confocal images of dissociated neurons expressing the constructs to demonstrate that the probes are trafficked well to the membrane?

2) Could the authors provide data (which should be on hand) demonstrating that Cl-out does not effect the baseline input resistance of neurons nor their resting E_{GABA} when not activated with light?

3) Using gtACR1 instead of gtACR2 could conceivably generate a Cl-extruder which only requires a single green excitation wavelength to activate both Arch and the gtACR. This might generate a tool which would be easier to use, perhaps mention this in the discussion?

4) After Cl-out4 was demonstrated to be a considerably more efficacious Cl-extruder than Cl-out1 in Figure 3, why was Cl-out1 chosen over Cl-out4 for the slice experiments?

5) Ref 27, is not appropriate in the context, as it is referring to GABAB and presynaptic effects.

6) For ease of understanding top of page 5, 3rd line should read 'We then investigated whether co-expression with the optogenetic proton...'

7) Page 7, Could the authors be more specific about where extracellular activity was recorded from in the hippocampal formation?

8) Bottom of page 8 with reference to Cl- imaging, the authors could consider also referencing recent tools for imaging Cl- in neurons specifically ie Raimondo et al 2013, and/or Paredes et al 2016.

9) The authors should also clearly state what the light power delivered at the specimen by the epi-fluorescence at 460 nm +/- 20 nm is.

10) Cl-out should represent potent neural silencing, could the authors comment on why they did not achieve complete silencing of spiking during light delivery - is this due to sparse/ low expression / poor light delivery?

A small tip, if the text could be justified (one click in word) and line numbers added, this makes the reviewing process more pleasant.

In summary, with some polish and a small amount of additional data this is certainly worthy of publication.

Reviewer #2 (Remarks to the Author):

In this manuscript the authors present a novel optogenetic method that provides the possibility to induce chloride extrusion. The method basic idea relies on the combination of two optogenetically active proteins: one induces membrane hyperpolarization while the other opens a chloride permeable channel. Prolonged outward directed flux of chloride under hyperpolarised conditions results in a reduction of intracellular chloride concentration leading to a transiently more hyperpolarised reversal potential for GABAA receptors.

While this manuscript reports an ingenious design with large potential scope in neuroscience there are a number of issues that would need to be resolved.

- 1) A major concern in this study is that the strategy precludes the possibility to directly assess changes in Egaba and alternative indirect estimations are needed as e.g. shift in the amplitude of muscimol induced IPSPs. These are though not only sensitive to changes in Egaba but also on changes in GABAA conductance that can be initiated by a number of alternative mechanisms. Control experiments should be performed to show that changes in IPSPs amplitude are not derived by changes in conductance.
- 2) GABAA receptors are also permeable for bicarbonate ions and as such Egaba is sensitive to changes in intracellular pH. Control experiments are needed to show that the eluded changes in Egaba are not caused by change in intracellular pH.
- 3) Only the results in figure 1 shows experiments that estimate Egaba the rest of the study uses incorrectly the term Egaba and intracellular chloride changes to describe changes in the amplitude of muscimol evoked IPSPs.
- 4) The effect of illumination of Cl-out1 co-opsin on VU0463271 induced spontaneous network activity as well as hyperexcitability could be mediated by the hyperpolarizing action of Arch. Control experiments using only Arch activation should be shown in fig 5.
- 5) Similarly for Fig 6

- 6) According to figure 3 the most potent construct is Cl- out 4. Why was the study performed with Cl-out-1 the weakest of the three functional constructs?
- 7) According to figure 4 and experiments showing the effect of VU0463271 on Egaba chloride extrusion is fully functional in the recorded neurons. However the responses shown in figure 2C shows depolarizing responses evoked by muscimol.

Minor comments:

- 1) The text is a bit messy and not precise. E.g. it is not clear what opsins are referred to in different paragraphs. This should be better specified. The text describing the recovery of depolarising IPSP after Cl-out1 activation is in a paragraph describing the results of fig 3 but based on the experiments shown in fig 4.
- 2) Figure 3 is not consistent: it shows a detailed characterisation of Cl-out 4 but not of Cl-out1. Although this manuscript is primarily base on the work on Cl-out1.
- 3) In general the manuscript would benefit from a structure where an initial assessment of the optimal configuration would be made. Then performing the rest of the study with the optimal construct.
- 4) A figure showing representative ramps for estimation of Egaba are missing in figure 1.
- 5) Also an experiment showing the absence of effect of Arch activation on GABAA conductance would be desirable.

Reviewer #3 (Remarks to the Author):

The authors describe an optogenetic approach to selectively reducing intracellular chloride concentration by combining two previously reported opsins: a blue light-activated anion channel (ChloC) and a green light-activated proton pump (Arch). The tool fills a niche allowing rapid and reversible decreases of intracellular chloride ions. While the effects of activating a chloride channel depend on the potential, co-activation of a proton pump provides a hyperpolarizing driving force to push chloride out. They then validate this strategy in cultured neurons and acute slice, showing that co-activation of the two options can mostly reverse the increased spiking caused by inhibiting KCC2 with VU0463271.

The combinatorial use of these two opsins for this purpose is interesting and appears to be novel. Although there are potential complications of co-expression of the opsin genes as well as co-delivery of blue and green light for activation, the reported approach should be valuable for a number of experiments, especially in vitro, where selectively decreasing intracellular chloride

would be informative. Experimental design, statistical analysis, and references look reasonable. I recommend publication with minimal revision.

We would like to thank the reviewers for their input. We were particularly pleased that they all clearly shared our enthusiasm for this new technology and were persuaded as to the potential usage in the neuroscientific community. We give our responses to the individual points made by the reviewers (*Reviewers' comments are italicized*). We have also added a few additional sentences to conform with the *Nature Communications* checklist.

Reviewer #1

Major comment...it is important that the authors convincingly show this tool working as a Cl- extruder in more intact preparations (ie acute slices) I remain unconvinced that the presented effects (ie Fig 5 and 6) on network activity are definitely due to changes in Cl- concentration and GABAergic signaling rather than direct voltage (Arch) or shunting (Chloc) inhibitory effects. Ie the effects described on epileptiform activity and spike timing relative to the dominant field oscillation (particularly during light illumination) could possibly be explained purely by the hyperpolarizing effect of Arch activation alone.

We now provide Arch and ChloC activation data also using VU (p8, line 9, and p9, line 10). We illustrate this data with two new figures (Figure 6 and Supplementary figure 5).

1) Use a Cl-out 'priming' approach. ie analyze evoked and spontaneous activity during the 5s that the light is off (ie following the 25s light activation).

This is in fact exactly what we did do for the stimulation data and the spike-timing analysis. We clarify the text regarding this point (p7, line 31).

It is worth noting that the authors state on pg 7 'Once the illumination was terminated, the hyperexcitable state was re-established, but not instantly, indicating that the inhibitory benefit of Cl-out far outlasts its activation, consistent with our cellular measures' - this is promising but no data is presented to support this claim.

In fact one trace in the original manuscript did illustrate this point (figure 5B), but we accept that this required further elaboration. We now include two extra figures (Figure 6B and Supplementary figure 4) addressing this point. We add a sentence with reference to these figures on p8, line 21.

2) It would be useful (but not essential) to perform some gramicidin perforated patch-clamp recordings in acute slices and directly demonstrate that a Cl-out is able to lower intraneuronal Cl-.

Perforated patch-clamp recordings are very difficult in brain slices, particularly in aging animals. We tried to do these, but the postdoc who was the expert in these recordings and the lead author of this study, has now moved to Oxford, so we had limited opportunities to pursue this – she did come back to Newcastle to collect the various other data requested, but these were the lowest priority (prompted by the reviewer's comment that these were "*not essential*"), and we were unable to tick this box. Having said that, it is quite clear that both opsins are operational in whole cell recordings (see Figure 2B) in brain slices.

Minor comments:

1) The Cl-out constructs are relatively large, and could conceivably result in cell trafficking / aggregation issues. If available could the authors provide confocal images of dissociated neurons expressing the constructs to demonstrate that the probes are trafficked well to the membrane?

We have made images of Clout 1 and Clout 4, which are now included as supplementary figure 2. We reference this at the top of page 6 (line 2).

2) Could the authors provide data (which should be on hand) demonstrating that Cl-out does not affect the baseline input resistance of neurons nor their resting E_{GABA} when not activated with light?

We provide the statistics to show that Cl-out does not affect these cellular properties unless activated by light (p 6, line 29f).

*3) Using *gtACR1* instead of *gtACR2* could conceivably generate a Cl-extruder which only requires a single green excitation wavelength to activate both Arch and the *gtACR*. This might generate a tool which would be easier to use, perhaps mention this in the discussion?*

This is in fact something we had considered, but it is not as straightforward as it might seem. ACR1 is activated by the same wavelength light but has slower kinetics, meaning that when the light is turned off, the ArchT effect stops but the ACR1 remain opens, and this might drive Cl⁻ on the other direction. We have added a sentence to this effect in the discussion (p10, line 10).

4) After Cl-out4 was demonstrated to be a considerably more efficacious Cl-extruder than Cl-out1 in Figure 3, why was Cl-out1 chosen over Cl-out4 for the slice experiments?

In fact Cl-out4 appears to be only marginally more efficacious than Cl-out1, although this difference was significant with relatively low sample sizes (n = 7 for both). We fear we may have inadvertently given the impression it was a bigger difference by the way we presented the data in figure 3, and we present a different version now, along with some additional statistical information.

Having said that, ideally we would indeed have used Cl-out4 for the network assays, but unfortunately the viral transfection using Cl-out4 was rather variable, and generally not good. Viral transfection with Cl-out1 on the other hand was much better. The most likely explanation for the expression patterns, we believe, is that we used two different AAV serotypes – Cl-out1 used AAV5, whereas Cl-out4 used AAV2. Given that expression was very good in cultures for Cl-outs 1,2 and 4 (3 was less good), when transfection was done by electroporation of the plasmids, we think that is probably unrelated to the actual Cl-out sequences. We include a short statement regarding the difference in viral expression patterns (page 7 line 19).

5) Ref 27, is not appropriate in the context, as it is referring to GABAB and presynaptic effects.

We have removed this reference. Please note that for some quirk of Endnote, in the version showing the tracked changes, this reference continues to exist in the reference list! It doesn't exist in the version with accepted changes.

6) For ease of understanding top of page 5, 3rd line should read 'We then investigated whether co-expression with the optogenetic proton...'

Actually, we disagree with this. The "co-expression" is not what provides the voltage clamp, it is the Archaeorhodopsin protein. No change made.

7) Page 7, Could the authors be more specific about where extracellular activity was recorded from in the hippocampal formation?

CA1 pyramidal cell layer. We have included this information in the results (p7, line 25) and methods sections (p19, line 5).

8) Bottom of page 8 with reference to Cl⁻ imaging, the authors could consider also referencing recent tools for imaging Cl⁻ in neurons specifically ie Raimondo et al 2013, and/or Paredes et al 2016.

Included (page 10 line 15, references 45-46).

9) The authors should also clearly state what the light power delivered at the specimen by the epi-fluorescence at 460 nm +/- 20 nm is.

Included (page 19 line 13).

10) Cl-out should represent potent neural silencing, could the authors comment on why they did not achieve complete silencing of spiking during light delivery - is this due to sparse/ low expression / poor light delivery?

Several factors may contribute. First, the illumination period does include short windows when the light is off, and when some spiking might occur. Note also that if E_{Cl} is very high, as is the case initially when bathed in VU, then the combined co-opsin activation may initially be relatively depolarising (see top trace in Figure 4A). Finally, the expression may not be optimised yet.

A small tip, if the text could be justified (one click in word) and line numbers added, this makes the reviewing process more pleasant.

Done!

Reviewer #2

1) A major concern in this study is that the strategy precludes the possibility to directly assess changes in E_{GABA} and alternative indirect estimations are needed as e.g. shift in the amplitude of muscimol induced IPSPs. These are though not only sensitive to changes in E_{GABA} but also on changes in GABAA conductance that can be initiated by a number of alternative mechanisms. Control experiments should be performed to show that changes in IPSPs amplitude are not derived by changes in conductance.

The example shown in Figure 2C could conceivably have arisen from a change in conductance, but there were other examples where the sign of the events switched from depolarising to hyperpolarising (see for instance the lower trace in Figure 4A (VU + bumetanide)). This could only arise from a shift in the Erev relative to resting E_m , and no amount of conductance change could do this. In the interest of addressing this point more directly, however, we present new data where we performed co-opsin activation when holding the cells in voltage-clamp, and these also confirm that the conductance is unaffected (supplementary figure 1). We have added text explaining these points on page 6 lines 9-15.

2) GABAA receptors are also permeable for bicarbonate ions and as such E_{GABA} is sensitive to changes in intracellular pH. Control experiments are needed to show that the eluded changes in E_{GABA} are not caused by change in intracellular pH.

GABA_A receptors are indeed permeable to bicarbonate, but it is far less so than for Cl (about 3-4 times less so), meaning that the bicarbonate reversal potential is much less important in dictating E_{GABA} than is the chloride reversal potential. Furthermore, intracellular pH tends to be buffered through the action of carbonic anhydrase, meaning that in most neurons, pH tends to be quite stable. This has been well studied with respect to bicarbonate / chloride exchange through GABAA receptors (see multiple publications from Kai Kaila's group). Putting those issues to one side, even if there were any change in pH, then the most likely direction of change would be for the intracellular pH to increase relative to the external pH, because of protons being pumped out of the cell by the Arch component of Cl-out. This though would cause a fractional positive shift in the bicarbonate reversal potential, which is the opposite to what we see in E_{GABA} . Thus, if bicarbonate were having any effect, if anything it would be to marginally occlude the chloride shift. In short, the mechanism the reviewer appears to be proposing is in fact opposite to the one we measure. The fact also that Arch activation alone does not affect cellular or network activity in the way Cl-out does (see various experimental data from our 2015 J.Neurosci. paper and in this study, including the extra recordings we have added here), supports this argument.

We recognise that this is actually an important point to discuss, and so we have added a small section to this effect in the discussion (page 9 line 24). But, given the difficulty of examining this issue experimentally (pH imaging is difficult given that we are dealing with other light sensitive proteins, and we also need to be non-invasive with our electrophysiology), the only additional data we provide on this point are the new recordings using just Archhodopsin activation in VU.

3) Only the results in figure 1 shows experiments that estimate E_{GABA} the rest of the study uses incorrectly the term E_{GABA} and intracellular chloride changes to describe changes in the amplitude of muscimol evoked IPSPs.

We accept that the later measures are all indirect. However, having now tightened the manuscript by (1) demonstrating a lack of change in conductance, and (2) arguing that if there are changes in bicarbonate, these are likely to have the opposite effect on E_{GABA} (see points 1 and 2 above), we would argue that these are indeed the true causes of the experimental changes we show in later figures. We feel that these terms are rather helpful "short-hand" descriptors, and we would prefer keep the text largely as it is.

4) The effect of illumination of Cl-out1 co-opsin on VU0463271 induced spontaneous network activity as well as hyperexcitability could be mediated by the hyperpolarizing action of Arch. Control experiments using only Arch activation should be shown in fig 5.

We have added controls to show that Arch does not cause these changes. See responses to reviewer 1 above.

5) Similarly for Fig 6

See point above.

6) According to figure 3 the most potent construct is Cl- out 4. Why was the study performed with Cl-out-1 the weakest of the three functional constructs?

Reviewer 1 also made the same point (see point 4) – please see our response above.

7) According to figure 4 and experiments showing the effect of VU0463271 on Egaba chloride extrusion is fully functional in the recorded neurons. However the responses shown in figure 2C shows depolarizing responses evoked by muscimol.

In fact, for the purposes of illustrating the effect, we actually chose two example recordings from tissue being bathed in VU0463271. We make this clear in the figure legend.

Minor comments:

1) The text is a bit messy and not precise. E.g. it is not clear what opsins are referred to in different paragraphs. This should be better specified. The text describing the recovery of depolarising IPSP after Cl-out1 activation is in a paragraph describing the results of fig 3 but based on the experiments shown in fig 4.

We have corrected these. The paragraph referred to simply segued from talking about figure 3 into describing figure 4, but we have separated these into two paragraphs, and trust that this helps matters.

2) Figure 3 is not consistent: it shows a detailed characterisation of Cl-out 4 but not of Cl-out1. Although this manuscript is primarily base on the work on Cl-out1.

We accept this, but as we point out in our other comments, the functional mechanisms of the two proteins are effectively the same, and they are perhaps not as dissimilar as the reviewer may have understood from the layout of figure 3 previously (we have altered this). We are not clear whether the reviewer is suggesting repeating all the experiments also for Cl-out1, but we feel this would unnecessarily slow the publication of this new technological development, for little extra benefit.

3) In general the manuscript would benefit from a structure where an initial assessment of the optimal configuration would be made. Then performing the rest of the study with the optimal construct.

We accept that stylistically, this may represent a marginal improvement, but at great extra effort, and one which also may not be perfect, because (a) it misrepresents the way the study was conducted, (b) it might result in failing to present certain information, and (c) it would delay publication substantially. One way to achieve this neat layout might be to remove the data about Cl-out4, since we were unable to fully assess the network effects, but we see no benefit in removing data. The Cl-out4 construct does appear to be marginally better in the culture assays, but sadly its viral vector did not work as well as Cl-out1's. So currently there is not a single "optimal" strategy. We therefore feel it is in the best interest for the scientific community to present the data in the way we have.

4) A figure showing representative ramps for estimation of Egaba are missing in figure 1.

We show the ramps in supplementary figure 1.

5) Also an experiment showing the absence of effect of Arch activation on GABAA conductance would be desirable.

We did not feel this justified the use of extra animals – ArchT does not change the amplitude of the muscimol event (Figure 3) or the shape (although we grant that the figure that would show that is rather small (Figure 2Cii) so the conductance can reasonably be presumed to be unaffected.

Reviewer #3

No substantive revisions suggested. We thank this reviewer also for the comment that the importance of this work merits "*publication with minimal revision*" in the interest of getting this out for the scientific community in a timely manner.

Reviewers' Comments

Reviewer #1 (Remarks to the Author):

This is an excellent manuscript. It turns out that my most substantive concern appears to be a misunderstanding about how the acute experiments were performed, I am now convinced by the value and performance of the CI-out strategy and thoroughly recommend the manuscript for publication. I request some very minor changes which do not require re-review on my behalf

1) At the risk of being pedantic for the benefit of the reader it could still be made clearer that in the acute experiments the electrical stimulation occurs when the light is off. For example in Figure 5 neither in the figure itself not in the legend does it say that the electrical stimulation and traces used for analysis are performed in the dark period of the illumination cycle. Indeed the zoomed in traces are presented in cyan with the caption VUO + light (which immediately makes one assume that the trace was recorded in the presence of light - esp as the light illumination bars above the main trace are also in cyan). Could I suggest the caption read something like VUO + light primed or VUO + CI-out primed - and the trace not be in cyan? The reader should be able to look at the figure and immediately understand how the experiment was setup. The same rationale applies to Figure 6 and 7. The new data in Figure 6 is fabulous and really adds to the strength of the paper / is useful to the field in terms of providing a comparison between silencing constructs.

2) In Supplementary Figure 2 showing confocal images of neurons expressing CI-out1 and CI-out4, although it is clear that the constructs are trafficked to the membrane, it is also quite obvious that significant intracellular aggregation is occurring, which may just be result of over expression. No changes requested.

Reviewer #2 (Remarks to the Author):

I have only a few minor comments that would still need attention otherwise the authors has addressed satisfactorily my concerns:

Minor comments:

Concerning point 3: I understand very well that naming Egaba changes as “short-hand” descriptors is helpful. But this is anyhow simply just incorrect and misleading for what it is actually measured. The Authors should address this changes in a correct manner and explain to the reader what it could imply e.g. changes in Egaba.

In supplementary figure 2 it is claimed to show membrane expression. This is not correct. They show distribution of the expressed proteins. To show membrane expression would need e.g. biotinylation or other methods.

Andrew Trevelyan
Institute of Neuroscience,
University of Newcastle,
Medical School,
Framlington Place,
Newcastle upon Tyne,
NE2 4HH.
UK

Tel (w): +44-191-222-5732

andrew.trevelyan@ncl.ac.uk

2nd September, 2016

Dear Dr Wright,

Thank you for overseeing the editorial process. We are delighted that the manuscript has now been accepted for publication and we look forward to seeing it online very soon.

The revised manuscript does not have the changes tracked, so I am just providing a list below, to indicate what specific changes we've made regarding the new comments from the reviewers

Reviewer 1

1. We have made changes to figures 5-7 as suggested, including the word “primed” where appropriate and also a small inset in figure 5, and have also added a line in the fig legend of figure 5 to make absolutely explicit that the electrical stimulation is in the dark period of the light cycle.
2. We have slightly changed the figure legend – mainly in response also to the comment by reviewer 2, removing the phrase about membrane expression and saying instead that the protein is transported throughout the cell.

Reviewer 2

1. To cater to this reviewer's firm stance about E_{GABA} , we introduce the term $GABA-\Delta V$ to refer to the specific measurements of the change in membrane potential at the peak of the GABAergic event. We add a line in the Methods to introduce the term (p17 line 30).

Note though that there are other instances when the use of E_{GABA} is actually correct (eg for measures of the baseline E_{GABA} in cells expressing the opsins, and also where we are referring generically to the effect of Cl-out), and in those instances, we insist that the use of the term “ E_{GABA} ” is accurate, and should remain.

2. The point about membrane expression is that we do show it by another method that is far more powerful even than biotinylation, which is that we show that the opsins are

functional, and carry membrane currents. We have however changed the phrasing of the figure legend in Supplementary figure 2, as noted above (see point 2 to Reviewer 1).

Finally, the 2 line Editor's summary is perfect.

Yours sincerely,

Andrew Trevelyan.
MB. BCh. DPhil (Oxon), MA (Oxon)